# Experimental Study on the Evaluation and Influencing Factors on Individual’s Emergency Escape Capability in Subway Fire

**DOI:** 10.3390/ijerph181910203

**Published:** 2021-09-28

**Authors:** Na Chen, Ming Zhao, Kun Gao, Jun Zhao

**Affiliations:** School of Mechanics and Safety Engineering, Zhengzhou University, Zhengzhou 450001, China; mingzhao718@163.com (M.Z.); kungup@163.com (K.G.); zhaoj@zzu.edu.cn (J.Z.)

**Keywords:** subway fire, emergency escape capability, virtual reality, physiological experiment, DISC type

## Abstract

Studying an individual’s emergency escape capability and its influencing factors is of great practical significance for evacuation and escape in subway emergencies. Taking Zhengzhou Zijing Mountain Subway station as the prototype, and using VR technology, a virtual subway fire escape scene was built. Combined with the total escape time, the total contact time with fire, and the total contact time with smoke, it proposed a calculation formula on emergency escape capability. A total of 34 participants with equal gender distribution were recruited to carry out the virtual subway fire escape experiment, and participants’ physiological data (heart rate variability, skin conductance) were real-time recorded by ErgoLAB V3.0 throughout the whole experiment. The emergency escape capability of each participant was evaluated quantitatively, and the related influencing factors were analyzed. The results show that for the age ranges (19–22 years old) in the experiment, the emergency escape capability of women is significantly lower than that of men (*p* < 0.05); although there is no significance in emergency escape capability in DISC personality types (*p* > 0.05), the mean emergency escape capability of people with influence personality type is the worst, and that of people with compliance type is the best; during virtual fire escape vs. baseline, Mean_SC and Mean_HR both increased very significantly (all *p* < 0.01), and participants were under stress during their virtual fire escape. There is a significant negative correlation between emergency escape capability and LF_increase_rate (*p* < 0.05), and a remarkably significant negative correlation between emergency escape capability and LF/HF_increase_rate (*p* < 0.01); the greater the increase rate of LF or LF/HF, the smaller the emergency escape capability, with excessive stress probably not being conducive to emergency escape. There is a very significant negative correlation between an individual’s emergency escape capability and the degree of familiarity with the Zijing Mountain subway station (*p* < 0.01). The findings provide references and suggestions on the emergency management and emergency evacuation for government and subway departments.

## 1. Introduction

As an important part of a modern urban public transportation system, the subway is an effective tool to relieve traffic congestion. However, subways are a limited underground space with crowded occupants, who are characterized by high density, high mobility, and complex composition. Once a fire occurs, a large number of casualties and economic losses can be caused, such as the Daegu subway fire in South Korea in 2003 [1], which caused 198 persons dead, 146 persons injured, and 298 missing persons. Passengers are the main object of subway emergency evacuation and rescue, and are also the main body that affects subway safety. Due to their individual differences in physiology and psychology, they usually show different emergency behaviors and escape capability. Therefore, it is of great practical significance to evaluate passengers’ emergency escape capability and find out the relevant influencing factors, so as to put forward targeted measures to improve the overall efficiency of emergency evacuation and escape in a subway environment. At the same time, it is also of important reference significance for the emergency management and emergency training of the government and related departments of subway systems. This paper intends to carry out an experimental study on the evaluation of passengers’ emergency escape capability and their relevant influencing factors by taking Zhengzhou Zijing Mountain subway station as an example and prototype; a transfer station of Line 1 and Line 2, with an underground structure comprising four floors and a large flow of occupants.

### 1.1. Escape Performance and Escape Capability

Usually, in an evacuation model, data about the evacuee’s pre-reaction time, movement speed, escape behavior, exit choice, individual characteristics, the influence of obstacles in the moving path on them, and the escape time are often considered to be collected [2]. The common performance indicators to measure evacuees’ emergency escape capability are escape time, escape distance, exit choice, and wayfinding ability, etc. Cao et al. [3] compared the similarities and differences between active and passive exploration in a virtual museum, and found people spent less time and moved less distance under passive exploration compared to active exploration. Tucker et al. [4] found that providing information about obstacles could reduce the total evacuation time of crowds. Sharma et al. [5]’s experimental study showed that competitive behavior could lead to physical collisions and inefficient evacuation. However, Heliövaara et al. [6] studied evacuees’ exit choices under different behavioral goals and found that the evacuation time of the whole crowd would be shorter when the evacuees showed self-interested behavior rather than cooperative behavior. Lin et al. [7], through a virtual reality experiment, found that there was a significant interaction between the completeness of spatial knowledge and the pattern of crowd flow, which had a negative impact on participants’ evacuation time, evacuation distance, and speed. Li et al. [8]’s research showed that a fire environment increases evacuation time, limits activity range, and changes evacuation routes. Purser [9] found that smoke in the fire environment would cause blurred vision and affect escape efficiency. Meng et al. [10], through a virtual reality experiment, compared people’s evacuation behavior and response with or without fire and found the evacuation efficiency of people under the fire condition is lower than that without fire. Kobes et al. [11] conducted an unannounced fire drill study in a hotel and found that most participants escaped through the main exit when no smoke was perceptible, while most participants escaped through the fire exit when smoke blocked the route to the main exit. Xu et al. [12] built a smoke hazard evaluation model to determine the safest evacuation route in the virtual fire environment.

Due to the “3I” (Imagination, Interaction, Immersion) features, virtual reality (VR) technology has been widely applied in different fields, such as military [13,14], medical science [15], education and training [16,17], etc., which makes evacuation experiments more economical [18] and safe [12,19,20]. Meanwhile, VR has also been proven to be a powerful tool in evacuation research [21], especially in evacuation decision-making [22] and people’s evacuation behavior [23,24]. Therefore, the first research objective of this paper is using VR technology to build a VR fire escape-scene and emergency escape capability evaluation system in a subway fire emergency, taking Zhengzhou Zijing Mountain Subway station as the prototype. Combined with the escape time and the contact time with fire or smoke, we propose a calculation formula of emergency escape capability innovatively. In addition, we designed the experimental procedure and selected the experimental subjects to carry out the experiment and determine each participant’s emergency escape capability quantitatively.

### 1.2. Physiological Indicators, Personality Characteristics, and Escape Capability

From the view of personal characteristics, evacuation can be affected by age, gender, physical condition, physiological and psychological quality, personality type, etc. Zijlstra et al. [25] simulated the difference among different age groups during an evacuation and found that elderly people had a lower moving speed than young people. Other studies also showed that age could lead to cognitive differences, and the decision-making time of the elderly group was longer than that of the younger group [26]. Men and women perform differently during evacuation [27,28]. In general, men exhibited better wayfinding performance than women within the same age range [29]. Nikolai et al. [30], through a virtual reality experiment, found that the pre-reaction time of men was shorter than that of women. Walkowiak et al. [31] found that women were generally more stressed during wayfinding than men. Choi et al. [32] simulated the evacuation through Anylogic and found that the vulnerable occupants, such as the elderly, the disabled, the sight or hearing impaired, and the cognitive disordered, performed differently in perception, interpretation, decision-making, and mobility.

Previous studies have shown that certain physiological indicators of individuals will change regularly under stress in case of emergency. In the virtual experimental study of Tucker et al. [4], Meng et al. [10], and Chen et al. [33], it was found that the value of SC (skin conductance) and HR (heart rate) in the virtual fire scenario increased significantly compared with the baseline state. Nassef et al. [34] founded that the HRV (heart rate variability) of individuals decreased with the increase of workload. Meina et al.’s [35] study found that compared with the calm state, the HRV value of firefighters in the task stage decreased. Filaire et al. [36] found that individuals’ RMSSD (root mean square difference between the Successive NN intervals) and HF (high frequency) decreased significantly, and LF/HF (ratio of LF (low frequency) to HF) increased significantly in the emergency state compared with the calm state. Delaney et al. [37] found that the SDNN (standard deviation of normal to normal (NN) interval) and HF of individuals were significantly reduced under short-term stress. Hjortskov et al. [38] found that compared with the calm state, the LF/HF and blood pressure of individuals increased during the computer work. When an emergency occurs, certain emotions and behavior of an individual will also not be conducive to the evacuation; e.g., Stein et al. [39] found that high levels of anxiety might have a negative impact on individual performance. In addition, participants are more likely to exhibit behaviors that are detrimental to evacuation when choosing routes under stress induction, such as moving to more crowded exits [30]. Anxiety emotion caused by emergency may also cause people to make sub-optimal choices, leading to slower evacuation speed and a greater risk of injury or even death [4].

Evacuation could also be influenced by an individuals’ personality traits and emotions [40]. Alexia et al. [41] proposed a design framework for evacuation simulation, which combined the OCEAN (Openness, Conscientiousness, Extraversion, Agreeableness, Neuroticism) personality model and OCC (Ortony, Clore, Collins) emotion model. Liu et al. [42] used the Big Five personality traits to describe the characteristics of different agents, and simulated the crowd evacuation in a virtual fire supermarket, and found that appropriate emotion contagion could speed up the evacuation. Zhan et al. [43] used a simplified Chinese version of NEO-FFI (Neuroticism Extraversion Openness Five-Factor Inventory) to assess students’ personality. The choice of classroom exits could be influenced by students’ conscientiousness—highly conscientious students were more rational when making decisions. Chen et al. [44], using a virtual subway fire escape experiment to explore the correlation between escape time and temperament type, found that people with sanguine temperament and choleric temperament had a longer escape time. Therefore, the second research objective of this paper is based on the built virtual subway fire emergency escape system, combined with the experimental data of SC and HRV and the related emotion and personality scale, to study the impact of gender, DISC personality type, and physiological indicators, etc., on an individual’s emergency escape capability, so as to improve subway’s risk resisting ability and emergency evacuation efficiency, and to provide references and suggestions on emergency management and emergency evacuation for government and subway departments.

## 2. Materials and Methods 

### 2.1. Participants

A total of 34 undergraduates from Zhengzhou University, 17 men and 17 women, mean age = 20.32 ± 0.88 years, ranging from 19 to 22 years old, were recruited to carry out the experiment. All participants had normal or corrected-to-normal vision, without any color vision deficiencies [45]. In addition, all participants had no diabetes, hypertension, nor any known heart, cardiovascular, or respiratory diseases [46,47], and also no vigorous activities or caffeine consumption within 24 h before the experiment [47]. All participants participated in the experiment on a voluntary basis and were paid accordingly after completing the experiment. All participants filled in the informed consent before the formal experiment.

### 2.2. Apparatus

All software and hardware used in the experiment are shown in Table 1 and Table 2.

### 2.3. Experiment Design

#### 2.3.1. Prototype Selection and Scene Building

Here, we chose Zhengzhou Zijing Mountain subway station (see Figure 1; this map can be accessed at: https://lbs.amap.com/api/javascript-api/example/map/map-english/ (accessed on 23 September 2021)), which is the largest transfer subway station of Line 1 and Line 2 in Zhengzhou, China, with a high passenger flow every day, as the prototype to build a VR subway fire escape scene using SketchUp and Unity3D, with 1:1 scale of the real station [33] (see Figure 2). There are four floors in Zhengzhou Zijing Mountain subway station. Level-1 is the transfer floor, Level-2 is the concourse level, Level-3 is the platform floor of Line 1, and Level-4 is the platform floor of Line 2. 

#### 2.3.2. Virtual Scene Settings

(1)Settings of fire, smoke, illumination, and brightness

The fire and smoke in the virtual scene were set by the particle system of Unity3D. According to the collision mechanism, the system would automatically count the contact time of the virtual character with fire or smoke. Vilar et al. [48,49] found that people preferred brighter corridors, so the illumination was evenly set in the virtual scene so as to reduce the influence of brightness on results. Considering the effect of smoke in fire, the brightness of the environment light was set to 50% of the normal brightness. 

(2)Settings of the starting point and ending point of the escape

The whole fire escape process includes two stages: the train escape stage and the station escape stage. The whole escape begins at the fire occurring on the train of Line 2 in Level-4 and ends with the participant’s arrival at the ground exits of the subway station. In the train escape stage, when the fire first occurs on the moving train of Line 2, participants need to perform a series of actions to get off the train, such as reporting the fire to the train driver through the emergency communicator, trying to put out the fire with the fire extinguisher, and opening the train door with the emergency door opener after the train arrives at the subway station. When participants get off the train and enter the subway station, they will start their station escape stage and need to continue their station escape until they arrive at the ground exits of the subway station.

(3)Settings of items

Fire extinguishers, towels, and bottled water are set in the virtual scene. Participants can use fire extinguishers to put out the fire on their escape paths, or pick up the towels and bottled water to resist fire and smoke, all of which can reduce their total contact time with the fire and smoke, and consequently change their emergency escape capability score. Participants can also pass through the fire or smoke directly during the escape, but their contact time with fire or smoke will increase, and thus change their final emergency escape capability score.

(4)Settings of the moving speed of the virtual characters

In the real emergency escape, the moving speed of people is generally 0.1–1.98 m/s [50]. In the virtual reality scenes, Lin et al. [51] set the moving speed of the virtual character as a fixed value of 2.4 m/s, and Cosma et al. [52] set this speed as 1.2 m/s. In order to avoid motion sickness caused by moving too fast or slow, here we set a constant speed of 1.5 m/s through several trials, with an Xbox joystick in hand to move or make turns in the virtual subway fire escape. In addition, the social-force influence of the characters was ignored in this experiment, and no NPCs (non-player characters) were set in the virtual escape scene.

(5)Setting of the maximum escape time

Gamberini et al. [21] set a maximum time of 15 min in the virtual experiment. Considering the long duration of the virtual experiment on people’s adverse effects, such as motion sickness, etc., here we set the upper limit of the total escape time in the whole virtual fire escape as 10 min. When participants could not escape to the ground exits within 10 min, the escape automatically ended. 

(6)Setting of the virtual teaching and practice scene

In order to make participants familiar with the virtual experiment operation before the formal virtual fire escape experiment, we first developed a virtual teaching and practice scene, which can teach and practice participants’ movement, the use of fire extinguishers, and the use of the emergency door opener, etc. in the virtual environment (see Figure 3).

#### 2.3.3. Setting of the Formula on Emergency Escape Capability 

In order to quantify participants’ emergency escape capability in a subway fire escape, according to their total escape time, their total contact time with fire, and their total contact time with smoke in the whole virtual subway fire escape, the following emergency escape capability formula was proposed:(1)The formula of emergency escape capability
Score of emergency escape capability = 100 − D_t_ − D_f_ − D_s_(1)
where D_t_ is the score deduction related to the total escape time, which is the penalty points for total escape time exceeding 300 s. D_f_ is the score deduction related to the fire contact, which is the penalty points for contacting fire. D_s_ is the score deduction related to the smoke contact, which is the penalty points for contacting smoke.

In Equation (1), a point deduction system is adopted. The full score of the emergency escape capability is set to 100 points. According to the statistics, most of the deaths in fires are caused by smoke rather than open flames [53], also, considering the importance of time to escape, here we set the maximum proportions of D_t_, D_f_, D_s_ as 40, 20, and 40% respectively.

(2)The formula of D_t_—Formula of penalty points for escape overtime
(2)$$D_{t}=\frac{T_{t}-300}{300}\times 40\%\times 100,\;0\leq D_{t}\leq 40$$
where T_t_ is the total escape time. Through the preliminary experiment, participants’ average total escape time is 300 s more or less, so here we set 300 s as the baseline of escape time, which is more strict because it is shorter than the 6 min of safety evacuation time that is stipulated in Code for Design of Metro (GB 50157-2013), Standard for Fire Protection Design of Metro (GB51298-2018), Code for Safety Evacuation of Metro (GB/T 33668-2017), and NFPA 130- 2017 (Standard for Fixed Guideway Transit and Passenger Rail Systems). 

In Equation (2), when T_t_ < 300 s, D_t_ is negative, then force D_t_ = 0. Namely, when T_t_ ≤ 300 s, then D_t_ = 0, which demonstrates that this participant’s total escape time is shorter, and there is no penalty points for escape overtime for him. When T_t_ = 600 s (namely the 10 min upper limit of the total escape time in the whole virtual fire escape), D_t_ = 40. Namely, when 300 s < T_t_ ≤ 10 min, there will be penalty points for escape overtime for the participant, and the range of points penalty for overtime is 0 < D_t_ ≤ 40. 40% is the proportion of escape overtime deduction.


(3)The formula of D_f_—Formula of penalty points for contacting fire
(3)$$D_{f}=\left(\frac{T_{f}}{10}\right)\times 20\%\times 100,\;0\leq D_{f}\leq 20$$
where T_f_ is the total contact time with fire. 10 s is the maximum time set for the virtual character that can contact with fire in the system. If T_f_ > 10 s, the escape will automatically end. 20% is the proportion of points deducted for contacting fire.


(4)The formula of D_s_—Formula of penalty points for contacting smoke.
(4)$$D_{s}=\left(\frac{T_{s}}{20}\right)\times 40\%\times 100,\;0\leq D_{s}\leq 40$$
where T_s_ is the total contact time with smoke; 20 s is the maximum time set for the virtual character that can be in contact with smoke in the system. If T_s_ > 20 s, the escape will automatically end; 40% is the proportion of points deducted for contacting smoke.

In Equations (3) and (4), 10 s is the maximum time of contact with fire and 20 s is the maximum time of contact with smoke, and are relevant to the specific amount setting of fire and smoke in our virtual subway fire escape scene. Due to people being more likely to come into contact with smoke than with fire in a fire condition, the amount of fire set in our virtual subway fire escape scene was also less than the amount of smoke. In addition, considering it would be hard for people to stand too long in the high heat of fire and smoke, according to the preliminary experiment, participants’ total contact time with fire and smoke were both less than 10 s and 20 s, respectively, so here we set the maximum time of contacting with fire is 10 s, and the maximum time of contacting with smoke is 20 s. However, this value should be determined from numerical simulation, which is also the focus of our future study.

After each participant finished his or her escape, the virtual scene will automatically jump to the results interface (See Figure 4). 

### 2.4. Experiment Procedure

Before the formal experiment, all of the participants had to sign an informed consent, fill in their basic personal information (see Appendix A), such as gender, age, etc., and take a DISC personality type test via a mobile phone Wechat Applet of the DISC test. DISC (Dominance (D), Influence (I), Steadiness (S), Compliance (C)), was created by William Marston, and is used to determine the personality type of different people [54]. Then, participants needed to use Xbox joysticks to exercise the basic operations, such as moving, use of fire extinguishers, etc., in the virtual practice scene (see Figure 3). 

Formal experiment phase: After the virtual practice, participants wore a PPG and EDA sensor and stayed calm for 2 min (see Figure 5a). Then, participants put on an Oculus DK1 (see Figure 5b) in order to escape in the virtual subway fire scene (see Figure 5c), with a Microsoft Xbox Elite joystick in hand. Meanwhile, participants’ SC and HRV data were real-time recorded by ErgoLAB V3.0, in the 2 minutes’ baseline period and the whole virtual fire escape period (see Figure 5d). During the whole escape process, participants were not told to notice any evacuation signs. After the escape was finished, participants needed to fill in a post-questionnaire (see Appendix A), which included their sense of reality, immersion, and nervousness in the virtual fire escape scene; their VR experience, degree of familiarity with the Zijing Mountain subway station, etc.

## 3. Results and Discussion

Data were collected and initially processed by ErgoLAB (all the data, including the questionnaire data, can be obtained from the link below: https://pan.baidu.com/s/1cAbFGt23qqtvdChkyzyN2g (accessed on 23 September 2021); Code: 7872). The data was analyzed by SPSS 22.0.

### 3.1. Score of Participants’ Emergency Escape Capability

According to the formula description on emergency escape capability in Section 2.3.3 in this paper, when each participant finishes his or her virtual fire escape, the system will automatically show his or her detailed escape score (See Figure 4). 

Thirty-four participants’ score of emergency escape capability (denoted as S_c_ below) are plotted in Figure 6, in which the black points represent each participant’s S_c_, and the red line represents all the participants’ average S_c_.

Regarding the influencing factors on participants’ emergency escape capability in gender, DISC personality type, and physiological indicators, before the analysis and discussion on the experiment results, we proposed the following hypotheses:

**Hypotheses** **1.**
*Gender is a factor affecting an individual’s emergency escape capability. Within the same age range, men generally have better escape capability than women.*


**Hypotheses** **2.**
*DISC personality type is a factor affecting an individual’s emergency escape capability. People with an influence personality type could have the worst escape capability.*


**Hypotheses** **3.**
*An individual’s emergency escape capability can be reflected by certain physiological indicators’ changes—the greater the fluctuation of certain physiological indicators, the lower the ability to escape. The escape performance may decrease with excessive stress.*


### 3.2. Relationship between Emergency Escape Capability and Gender

In Figure 6, the S_c_ of men was numbered 1–17, and the S_c_ of women was numbered 18–34. Among them, a total of six men’s Sc were all lower than the average S_c_, while ten women’s S_c_ were lower than the average S_c_.

From the tests of normality and homogeneity of variance test in SPSS, the S_c_ in men and women all obeyed normal distribution and homogeneity of variance, so the independent sample T-test was used to determine the difference of S_c_ in gender (see Table 3). From Table 3, there is a significance in S_c_ in gender (*p* < 0.05), and the mean of S_c_ in women (75.37 ± 11.93) was significantly lower than that in men (83.79 ± 10.03). This reflects that within the age group (19–22 years old), men’s emergency escape capability was better than women’s. Gender is a factor affecting an individual’s emergency escape, which verifies Hypotheses 1 and is also consistent with the existing research [55].

### 3.3. Relationship between Emergency Escape Capability and DISC Personality Type

From the one-way ANOVA results in SPSS (see Table 4), there was no statistical significance in S_c_ in the four kinds of DISC personality types (*p* > 0.05).

Some studies have shown that people with a dominance personality type usually make quick decisions and often show high initiative and vitality; people with an influence personality type are often fickle and curious; people with a steadiness personality type are patient, loyal, and good at creating a stable and harmonious atmosphere, and their biggest motivation is the sense of safety; people with a compliance personality type are careful and prudent, and can think ahead and prevent problems [54]. According to this, we put forward the above Hypotheses 2, and considered people with influence personality type could have the worst escape capability because they are inconstant and easily influenced. From Figure 7 and Table 4, although there is no significance in Sc in DISC personality type, the mean of S_c_ in compliance personality type (83.09 ± 9.02) is the highest, and the mean of S_c_ in influence personality type (77.43 ± 9.42) is the lowest, which verifies Hypotheses 2. The reason may be that under the emergency stress, people with influence type are fickle and easily influenced by the surrounding environment, and have no their own opinion on the escape direction, while people with compliance type are more prudent at obtaining evacuation guiding information and can make correct decisions. This provides thoughts for targeted training on different personality types of occupants in the future.

### 3.4. Relationship between Emergency Escape Capability and Physiological Indicators

The SC and EDA data were processed by Gaussian filtering and low-pass filtering (both using default values) in ErgoLAB V3.0. The HRV data were processed by Wavelet denoising, high-pass filtering, low-pass filtering, and band-hinder filtering (all using default values) in ErgoLAB V3.0.

Previous relevant studies have shown that when people are under stress, their physiological indicators will change. Usually, their SC will increase [4,10]. Meanwhile, their SNS (sympathetic nervous system) is activated and PNS (parasympathetic nervous system) activity is inhibited [56], and the time domain parameters of HRV are generally shown as a decrease of SDNN [37,57,58], a decrease of RMMSD [36,57,58], and an increase of HR [59,60]; the frequency domain parameters of HRV are generally shown as an increase of LF [61], a decrease of HF [62], and an increase of LF/HF [63]. 

From the tests of normality and homogeneity of variance test in SPSS, the physiological data during virtual fire escape and baseline period all obeyed normal distribution and homogeneity of variance, so the paired-sample T-test was used to determine the changes of physiological data during virtual fire escape compared to the baseline period. From the paired-sample T-test results (see Table 5), during virtual fire escape vs. baseline, Mean_SC (the average value of SC) and Mean_HR (the average value of HR) both increased very significantly (both mean difference > 0, both *p* < 0.01, both d (effect size) > 0.8); SDNN, RMSSD, and LF/HF all decreased without significance (all mean difference < 0, all *p* > 0.05); LF, HF both increased without significance (both mean difference > 0, both *p* > 0.05). These indicated that participants were under stress during their virtual fire escape. This was also verified by the post-questionnaire, in which more than half of the participants felt nervous in the virtual fire escape. 

From the Tests of Normality in SPSS, the S_c_, Mean_SC_increase_rate, Mean_HR_increase_rate, SDNN_increase_rate, RMSSD_increase_rate, HF_increase_rate all obeyed normal distribution, so the Pearson correlation test was used to determine the relationship between S_c_ and each physiological indicator. From the Pearson correlation results (see Table 6), there was no correlation between S_c_ and Mean_SC_increase_rate, Mean_HR_increase_rate, SDNN_increase_rate, RMSSD_increase_rate, HF_increase_rate (all *p* > 0.05); while there is a significant negative correlation between S_c_ and LF_increase_rate (*p* < 0.05), and a remarkably significant negative correlation between S_c_ and LF/HF_increase_rate (*p* < 0.01). This indicates that the greater the increase rate of LF and LF/HF, the smaller the S_c_, while an increase of LF and LF/HF represents the person being in a stressful state, which may reflect that excessive stress is not conducive to emergency escape (which verifies Hypotheses 3). Therefore, during the whole emergency evacuation process, subway management departments should take various emergency management and emergency evacuation measures to maintain occupants’ emotional stability and avoid their excessive stress; for example, designing and setting effective dynamic flickering emergency evacuation guidance signs, ensuring power supply for emergency lighting, playing subway staffs’ roles as much as possible to guide occupants’ emergency evacuation, providing helpful evacuation information through broadcast, using laser-guided emergency evacuation technology and systems, etc.

### 3.5. Other Influencing Factors on Emergency Escape Capability

According to the post-questionnaire, 73.5% of the participants thought the virtual fire escape scenes were of a high degree of immersion; 73.5% of the participants had computer game experience; 32.4% of the participants had VR experience, which was mainly VR games or VR movies; 97.1% of the participants thought the operation in the virtual scene was general or not difficult or not difficult at all; 64.7% of the participants thought it was not difficult or not difficult at all to find the escape exits; 70.6% of the participants were unfamiliar or completely unfamiliar with the Zijing Mountain subway station. Here, a Spearman correlation test was used to determine the relationship between S_c_ and other influencing factors. From the Spearman correlation results (see Table 7), the immersion degree in the virtual fire escape scene, individuals’ computer gaming experience, VR experience, and difficulty in operation in the virtual scene, all had no impact on S_c_ (all *p* > 0.05), which indicated that the experimental design in this paper eliminates the influence of certain irrelevant variables on the experimental results well. Furthermore, from Table 7, there was no correlation between S_c_ and difficulty in finding the escape exits (*p* > 0.05), while there was a very significant negative correlation between S_c_ and the degree of familiarity with the Zijing Mountain subway station (*p* < 0.01), which indicates that the more familiar with the subway station, the higher the S_c_ was. This provides a way to improve an individual’s emergency escape capability by becoming familiar with the station consciously, including familiarity with various escape routes and emergency exits after entering the subway station, etc.

### 3.6. Limitations

(1)Virtual scene

The fire and smoke in the virtual scene were not numerically simulated, the pattern of which may not conform to the real condition and limit the immersion. In future, fire and smoke will be numerically simulated to provide a more realistic scene visually. In addition, there were no NPCs set in the virtual escape scene without considering the characters’ social force impact. More NPCs will be added to the virtual scene in future studies.

(2)The formula of emergency escape capability

In the real fire environment, there are various complex factors affecting the escape, and thus, it is difficult to fully evaluate an individual’s emergency escape capability. The formula of emergency escape capability in this paper is only applicable to this study’s virtual subway fire escape system and can provide a reference for similar research. Whether it is applicable to other system models remains to be further verified. Moreover, regarding the experimental results on influencing factors on an individual’s emergency escape capability, only one single prototype of Zijing Mountain subway station was used in this paper, which may reduce the reliability on the research results. Other prototype studies will be carried out to draw more credible conclusions in the future.

(3)Participants

Participants were all undergraduates (ranging from 19 to 22 years old) from Zhengzhou University in China; the sample size is a little small. Considering the prevalence of COVID-19 and even Delta variant everywhere, due to the limitations of the experimental conditions, we only chose undergraduates in our university as participants. However, choosing participants to be consistent with the actual composition of subway passengers would draw more credible conclusions. In our future studies, we will expand subject sample groups and size, and choose participants to be consistent with the real composition of subway occupants as much as possible. We will also explore other factors that affect an individual’s emergency escape capability, like different age groups, people with wheelchairs, hearing impaired, medication, substance use, tiredness, drunkenness, or people traveling in groups, or other individual characteristics, etc., in our future research.

## 4. Conclusions

In this paper, 34 participants’ emergency escape capability in a virtual subway fire were evaluated, and the related influencing factors were analyzed. For the age ranges (19–22 years old) in this experiment, the following conclusions were drawn: (1)A calculation formula of emergency escape capability is proposed, which can evaluate an individual’s emergency escape capability quantitatively in subway virtual fire escape.(2)Gender is a factor affecting an individual’s emergency escape; men’s emergency escape capability is better than women’s (*p* < 0.05).(3)There is no significance in emergency escape capability in DISC personality type (*p* > 0.05). Although there is no significance in emergency escape capability in DISC, the mean emergency escape capability in compliance personality type is the best, and the mean emergency escape capability in influence personality type is the worst. This provides thoughts for targeted training on different personality types of occupants in subways in the future.(4)During virtual fire escape vs. baseline, Mean_SC and Mean_HR both increased very significantly (all *p* < 0.01), which indicated that participants were under stress during their virtual fire escape. Moreover, there is a significant negative correlation between an individual’s emergency escape capability and LF_increase_rate (*p* < 0.05), and a very significant negative correlation between an individual’s emergency escape capability and LF/HF_increase_rate (*p* < 0.01). This indicates that the greater the increase rate of LF and LF/HF, the smaller the emergency escape capability, while an increase of LF and LF/HF represents the person being in a stressful state, which may reflect that excessive stress is not conducive to emergency escape. Therefore, during the whole emergency evacuation process, subway management departments should take various emergency management and emergency evacuation measures to maintain occupants’ emotional stability and avoid their excessive stress.(5)There is a very significant negative correlation between an individual’s emergency escape capability and the degree of familiarity with the Zijing Mountain subway station (*p* < 0.01), which indicates that the more familiar with the subway station, the higher the emergency escape capability is. This provides a way to improve an individual’s emergency escape capability by becoming familiar with the station consciously, including familiarity with various escape routes and emergency exits after entering the subway station, etc.

## Figures and Tables

**Figure 1 ijerph-18-10203-f001:**
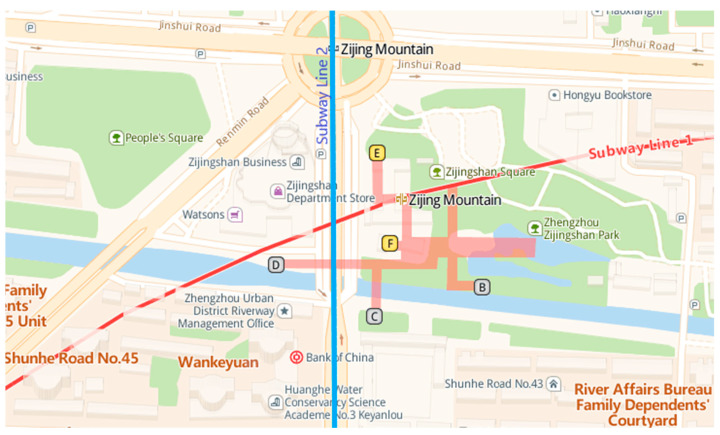
Geographical map of Zijing Mountain subway station in Zhengzhou, China.

**Figure 2 ijerph-18-10203-f002:**
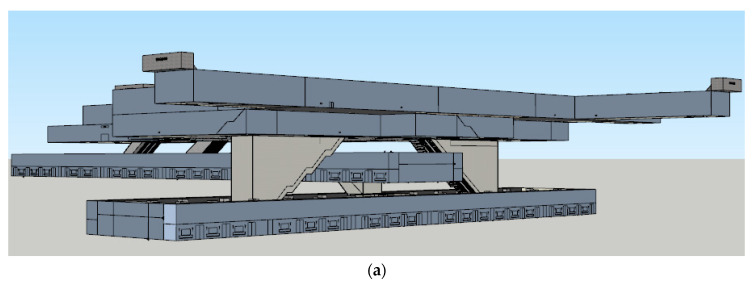
A 3D model screenshot of Zijing Mountain subway station in SketchUp, 1:1 scale of the real station. (**a**) The exterior view of the 3D model of Zijing Mountain subway station. (**b**) Guide Signs in the subway station. (**c**) The stairway in the subway. (**d**) The interior view of the subway car.

**Figure 3 ijerph-18-10203-f003:**
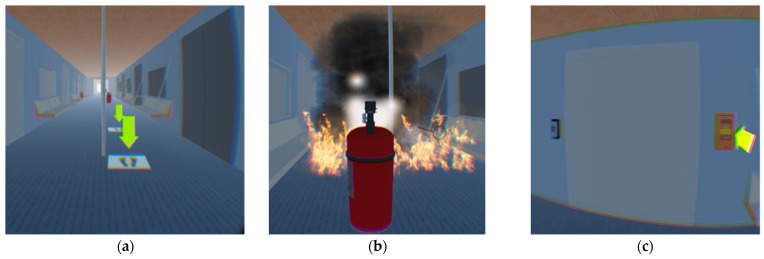
Screenshot of the teaching and practice scene. (**a**) Movement teaching and exercise. Participants need to move to the footprint under the yellow arrow with an Xbox joystick in their hands. (**b**) Teaching for use of the fire extinguishers. Participants need to learn how to use the Xbox joystick to put out the fire. (**c**) Teaching for use of the emergency door opener with the Xbox joystick in hands.

**Figure 4 ijerph-18-10203-f004:**
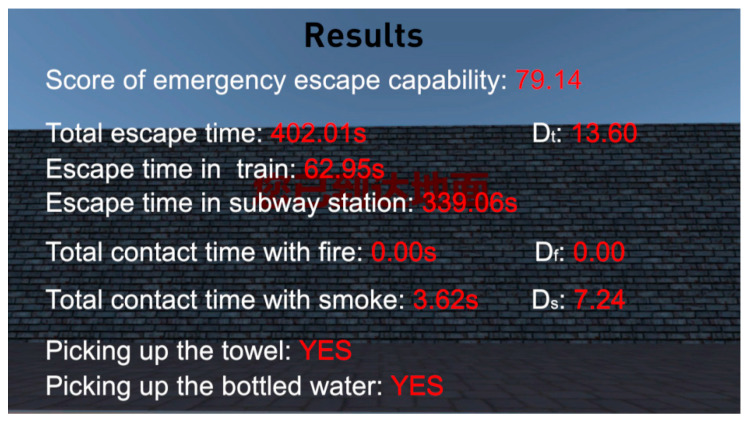
The results display interface after one of the participants finished his virtual escape experiment.

**Figure 5 ijerph-18-10203-f005:**
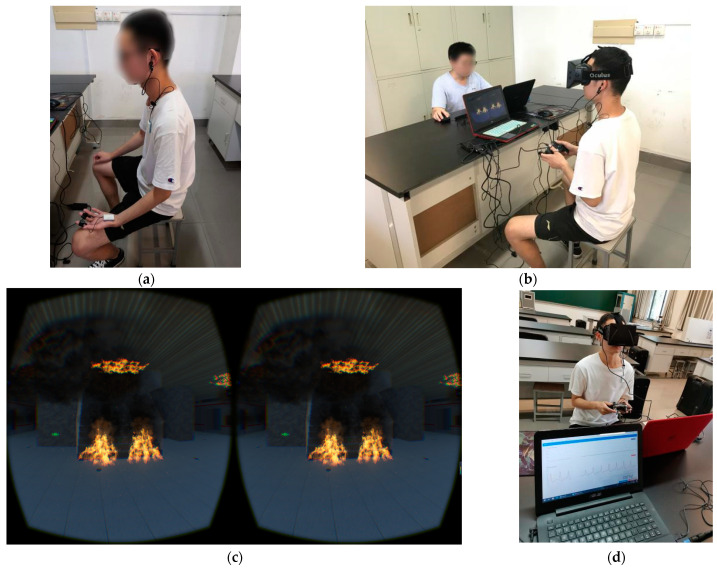
(**a**) Participants need to keep calm for 2 min in the baseline period, with EDA sensors on their index and middle fingers and PPG sensors in their earlobes. (**b**) Participants wore an Oculus DK1 HMD in the whole virtual fire escape, with a Microsoft Xbox Elite joystick in their hands. (**c**) The virtual subway fire escape scene. (**d**) Participants’ SC and HRV data are real-time recorded by ErgoLAB in the baseline period and the whole virtual fire escape period.

**Figure 6 ijerph-18-10203-f006:**
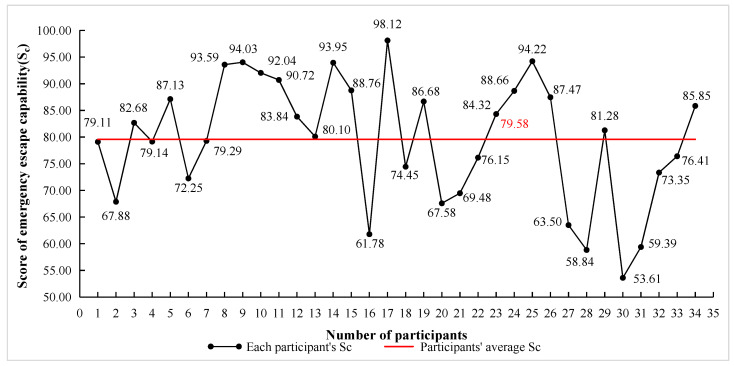
Participants’ score of emergency escape capability (S_c_).

**Figure 7 ijerph-18-10203-f007:**
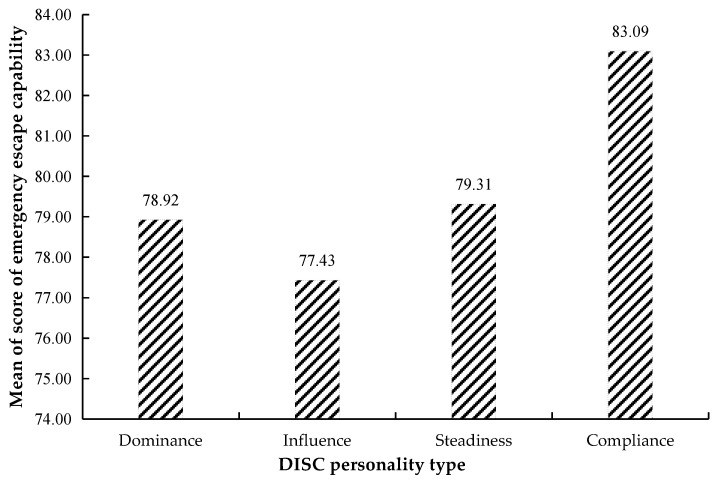
The mean score of emergency escape capability in the DISC personality type.

**Table 1 ijerph-18-10203-t001:** The software used in the experiment.

Name	Description
SketchUp (Version of 2015pro, Trimble, Sunnyvale, CA, USA)	The 3D subway VR model was built by it, according to the real Zijing Mountain subway station in Zhengzhou, China, with 1:1 scale
Unity3D (Version of 2018.2.10f1, Unity Technologies, San Francisco, CA, USA)	The built model was imported into it for the interactive setting of scene functions
Adobe Photoshop CS6 (64 Bit) (Adobe Systems Incorporated, San Jose, USA), Microsoft Paint3D (Microsoft Corporation, Redmond, WA, USA)	Some images were designed by them as textures in Unity3D
ErgoLAB V3.0 man-machine-environment synchronous cloud platform (Kingfar International Inc., Beijing, China)	Be used for real-time physiological data acquisition and processing, including ErgoLAB wearable wireless physiological recording module (PPG (photoplethysmography), and EDA (electrodermal activity))
IBM SPSS Statistic 22.0 (IBM Corporation, Armonk, NY, USA)	Be used for data analysis

**Table 2 ijerph-18-10203-t002:** The hardware used in the experiment.

Name	Description
Oculus Rift DK1 (Development Kit 1) ((c) 2013 Oculus VR, Inc., California, USA)	The virtual reality head-mounted display (HMD), with a resolution of 1280 × 800
Microsoft Xbox One Elite (Microsoft (China) Corporation, Beijing, China)	Participants can use it to move, make turns, or trigger various operations such as fire extinguisher, emergency communicator, emergency door opening device, etc., in the virtual subway fire escape scene
PPG (photoplethysmography) sensor (Kingfar International Inc., Beijing, China), EDA (electrodermal activity) sensor (Kingfar International Inc., Beijing, China)	The real-time physiological data acquisition device. The PPG sensor was clipped to participant’s earlobe to measure and record their HRV data, and the EDA sensor was tied to participant’s index and middle fingers, in good contact with the skin, to record their SC data

**Table 3 ijerph-18-10203-t003:** Independent Sample T-test of participants’ Sc in gender.

Gender	Number	Mean	SD (Std. Deviation)	Sig. (2-Tailed)
Man	17	83.79	10.03	0.033 *
Woman	17	75.37	11.93	

Note: * means the value of Sig. (2-tailed) is less than 0.05.

**Table 4 ijerph-18-10203-t004:** One-way ANOVA of participants’ S_c_ in DISC personality type.

DISC Personality Type	Number	Mean	SD	Sig. (2-Tailed)
Dominance	6	78.92	15.26	0.831
Influence	8	77.43	9.42	
Steadiness	13	79.31	13.19	
Compliance	7	83.09	9.02	

**Table 5 ijerph-18-10203-t005:** Paired-sample T-test of physiological indicators during escape vs. baseline.

Physiological Indicators during Escape vs. Baseline	Mean Difference	SD	d (Effect Size)	Sig. (2-Tailed)
Mean_SC during escape—Mean_SC baseline	2.08	2.56	0.81	0.000 **
Mean_HR during escape—Mean_HR baseline	5.94	6.96	0.85	0.000 **
SDNN during escape—SDNN baseline	−14.29	173.04	0.08	0.633
RMSSD during escape—RMSSD baseline	−5.01	219.98	0.02	0.895
LF during escape—LF baseline	1,578,315.61	9,400,937.52	0.17	0.335
HF during escape—HF baseline	6085.99	25,935.61	0.23	0.180
LF/HF during escape—LF/HF baseline	−0.756	17.21	0.04	0.799

Note: ** means the value of Sig. (2-tailed) is less than 0.01.

**Table 6 ijerph-18-10203-t006:** Correlation between Sc and physiological indicators.

		Mean_ SC_Increase_Rate	Mean_ HR_Increase_Rate	SDNN_Increase_Rate	RMSSD_Increase_Rate	LF_Increase_Rate	HF_Increase_Rate	LF/HF_Increase_Rate
S_c_	Pearson correlation	−0.109	−0.096	−0.241	−0.185	−0.422	−0.160	−0.485
	Sig. (2-tailed)	0.541	0.590	0.170	0.294	0.013 *	0.367	0.004 **

Note: The increase rate is the growth rate of the physiological indicator during virtual fire escape vs. Baseline; * means the value of Sig. (2-tailed) is less than 0.05; ** means the value of Sig. (2-tailed) is less than 0.01.

**Table 7 ijerph-18-10203-t007:** Correlation between Sc and other influencing factors.

		Immersion Degree in the Virtual Scene	Computer Gaming Experience	VR Experience	Difficulty in Operation	Difficulty in Finding Exits	Degree of Familiarity with Subway Station
S_c_	Spearman correlation	−0.173	0.044	−0.048	−0.290	−0.106	−0.446
	Sig. (2-tailed)	0.328	0.804	0.787	0.097	0.552	0.008 **

Note: ** means the value of Sig. (2-tailed) is less than 0.01.

## Data Availability

All the experiment data including the questionnaire data in this paper, can be obtained from the link below: https://pan.baidu.com/s/1cAbFGt23qqtvdChkyzyN2g (accessed on 23 September 2021); Code: 7872.

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
