# Peer review of "Experimental Study on the Evaluation and Influencing Factors on Individual’s Emergency Escape Capability in Subway Fire"

_ijerph, 2021, doi:10.3390/ijerph181910203_

Round 1
Reviewer 1 Report
In this paper, the virtual reality technology is designed to simulate the subway vehicle fire, and the stress physiological response characteristics of subway passengers during the fire period are analyzed by using the experimental method. The accurate test data are obtained and the statistical processing and impact factors analysis are carried out. The research methods are advanced and the technical route is reasonable. The research results are of some theoretical significance and have some reference value for further research in this field. Therefore, in my opinion,this manuscript can be accepted, if the author can revise the following suggestions:
- It is suggested to define all symbols used in the manuscript separately.
- Among the published papers, there are papers using Virtual Subway Fire Escape and individual physiological stress analysis, which title are Virtual scene of subway fire and experimental study on individual physiological stress and The Physiological Experimental Study on the Effect of Different Color of Safety Signs on a Virtual Subway Fire Escape—An Exploratory Case Study of Zijing Mountain Subway Statio It is necessary to explain the relationship between this paper and the previous related papers, and to cite the relevant papers published before in the references.
- In line 147, the author chooses the young people (19-22 years old) as the research object, which is inconsistent with the actual composition of subway passengers, so the reasons should be explained clearly.
- In line 234 and 235, the maximum allowable escape time is 10min, but in Code for design of Metro the allowable escape time is 6min, it is suggested to explain why the manuscript use 10min and the baseline of escape time is half of the maximum allowable escape time. Also in line 240 and 244, it is suggested to explain why the maximum time of contacting with fire is 10s and the maximum time of contacting with smoke is 20s.
- In line 256 to 258, has it been considered that if getting familiar with the system scene in advance will affect the impact of path familiarity on evacuation results?
- It is suggested to provide sources of the data cited in the manuscript.
- A sample questionnaire and experimental process record can be included.
- It is suggested to declare why SC and HR is the stress indicator and LF and LF/HF can reflect the excessive stress.
- The research result of this paper has guidance and reference value for emergency management and personnel guidance of subway fire, but it does not involve how to guide evacuation. It is suggested that the contents of analysis and discussion on the significance be added.
- Once again check throughout for language and grammar errors.
Reviewer 2 Report
Using VR in an experimental study on the evacuation capability in subway fire provides important information. To study mass evacuation scenarios is both difficult and expensive. The subject of the study is therefore important. Also seeing the group of escaping people as individuals provides a good view on the issue. The references were well chosen and the context description, such as game metrics were well and clearly written. Moreover, the pictures supported the description well. I recommend that this study would be published with minor changes. To be considered:
-This study was a preliminary attempt to design and develop an experiential model for evaluating individual´s escape capability, therefore the results are preliminary and further studies with bigger sample should be carried out
-The preliminary result with 34 students does not take into account that in a real emergency situation at the station there would be people with lowered escape capability (for instance people with wheel chairs, hearing impaired); people travelling in groups or other situational factors (medication, substance use, tiredness) that may affect the ability to escape.
For gender equality reasons consider also the use of words man and woman instead of male and female (note: female student, male passanger is correct).
Reviewer 3 Report
Using VR technology, a virtual subway fire escape scene taking Zhengzhou Zijing Mountain Subway station as the prototype is presented in the paper. Combined with the total escape time, the total contact time with fire, and the total contact time with smoke, it is proposed a calculation formula on emergency escape capability.
A total of 34 participants with equal gender distribution had been recruited to carry out the virtual subway fire escape experiment, and Participants’ physiological data (heart rate variability, skin conductance) were real-time recorded by ErgoLAB V3.0 in the whole experiment.
The emergency escape capability of each participant had been evaluated quantitatively, and the related influencing factors were analyzed.
These would provide references and suggestions on the emergency management and emergency evacuation of the government and subway departments.
It will be good to outline reasons for choosing concrete values of parameters in the formulas.
Round 2
Reviewer 1 Report
The authors carefully replied to each review comment and addressed with added references.
The responses were adequate and reasonable.
Agree to publish.